# Daily-Life Walking Speed, Quality and Quantity Derived from a Wrist Motion Sensor: Large-Scale Normative Data for Middle-Aged and Older Adults

**DOI:** 10.3390/s24165159

**Published:** 2024-08-10

**Authors:** Lloyd L. Y. Chan, Stephen R. Lord, Matthew A. Brodie

**Affiliations:** 1Neuroscience Research Australia, Sydney, NSW 2031, Australia; l.chan@neura.edu.au (L.L.Y.C.); s.lord@neura.edu.au (S.R.L.); 2School of Health Sciences, University of New South Wales, Sydney, NSW 2052, Australia; 3School of Population Health, University of New South Wales, Sydney, NSW 2052, Australia; 4Graduate School of Biomedical Engineering, University of New South Wales, Sydney, NSW 2052, Australia

**Keywords:** gait speed, demographics, normative values, accelerometer

## Abstract

Walking is crucial for independence and quality of life. This study leverages wrist-worn sensor data from UK Biobank participants to establish normative daily-life walking data, stratified by age and sex, to provide benchmarks for research and clinical practice. The Watch Walk digital biomarkers were developed, validated, and applied to 92,022 participants aged 45–79 who wore a wrist sensor for at least three days. Normative data were collected for daily-life walking speed, step-time variability, step count, and 17 other gait and sleep biomarkers. Test–retest reliability was calculated, and associations with sex, age, self-reported walking pace, and mobility problems were examined. Population mean maximal and usual walking speeds were 1.49 and 1.15 m/s, respectively. The daily step count was 7749 steps, and step regularity was 65%. Women walked more regularly but slower than men. Walking speed, step count, longest walk duration, and step regularity decreased with age. Walking speed is associated with sex, age, self-reported pace, and mobility problems. Test–retest reliability was good to excellent (ICC ≥ 0.80). This study provides large-scale normative data and benchmarks for wrist-sensor-derived digital gait and sleep biomarkers from real-world data for future research and clinical applications.

## 1. Introduction

Good quality daily-life walking is crucial for functional independence, an indicator of quality of life and a predictor of multiple advantageous health outcomes in older people. Wearable sensor-based gait assessments enable objective and direct assessment of walking performance in real-world settings. These digital gait biomarkers have demonstrated superior accuracy in predicting injurious falls [1,2] and identifying neurological and mental health conditions [3,4] when compared to conventional set-distance clinical gait assessments.

Establishing normative values for daily-life walking would provide important benchmarks to identify impairments in individuals relative to their age and sex-matched peers. In addition, such data provide evidence-based rehabilitation goal setting for individuals and benchmarks at public health policy levels. Normative values of standardized gait speed in middle-to-older age adults [5,6] are available, and have facilitated the interpretation of walking measurements in research studies and clinical practice. However, these normative data are not directly translatable to real-world gait performances, as several studies have found that these measures are only moderately correlated. [7,8,9].

Motion sensors have often been placed on the wrist, waist, thigh, and ankle. Of these, the wrist is arguably the preferred location for long-term continuous sensor attachment, as it is the most convenient and comfortable for the wearer [10,11,12,13]. Indeed, smartwatches and bands are now widely used in the general community. Wrist-worn accelerometers have now been used in recent large-scale population studies, including the UK Biobank [14], National Health and Nutritional Examination Survey (NHANES) [15], the Rotterdam study [16], and the Irish Longitudinal Study on Ageing (TILDA) [17]. However, as of now, normative values for daily-life walking performance based on wrist-worn sensors have not been derived.

To address this gap, we analysed one-week wrist sensor data from 106,053 UK Biobank participants using the Watch Walk method [18]. To summarise, this method captures wrist-worn acceleration signals to identify walking patterns based on hand movements (e.g., arm swing, hands in pockets) using machine learning with Support Vector Machine (SVM) classification. We examined 20 daily-life walk- and sleep-related biomarkers, including walking quality and quantity quantified through frequency-domain and peak detection analyses, and walking speed estimated using SVM regression. This study aimed to derive normative values for walking parameters for the cohort, and to compare these measures between men and women and across different age groups.

## 2. Materials and Methods

### 2.1. Development of the Watch Walk Digital Biomarkers

We recruited 101 participants aged 19 to 91 (mean: 47, standard deviation, SD: 18) (67% female) from Sydney, Australia and Hong Kong, China. All participants reported no osteoporosis, neurological impairment, recent fractures, recent joint replacements or existing conditions that restrict exercise tolerance. Written consent was obtained before data collection, and ethics approvals were obtained from the University of New South Wales and Hong Kong Caritas Institute of Higher Education Human Research Ethics Committees. An accelerometer (AX3 data logger, Axivity Ltd., Newcastle upon Tyne, UK) was positioned over the participant’s dominant wrist and was videotaped during the data collection. They conducted thirty minutes of daily activities and walked on an electronic walkway (GAITRite, CIR System Inc., Franklin, NJ, USA) with six different hand positions, including walking (1) with arm swing, (2) with hands in pockets, (3) with hands held static in front of the body (texting), (4) with the dominant hand held next to the ear (phone call), (5) while carrying a bag over the shoulder, and (6) while carrying a briefcase/grocery bag at slow, usual and fast paces.

We referenced the video recordings for the synchronization and annotation of the accelerometric signals and ground-truth electronic walkway measurements, including walking speed and step time. We derived the sample level Euclidean norm from the annotated accelerometric signals and filtered it with a Butterworth bandpass filter (5th order, 0.25 Hz low cutoff frequency, 2.5 Hz high cutoff frequency, and 100 Hz sampling frequency). The filtered signals were then segmented into non-overlapping 4-second windows. Fifty-four features were extracted for training and validating a two-stage multiclass support vector machine (SVM) classification model with ten fold validation at the participant level. In the first stage, the 4-second window was first classified into “walks with arm swing”, “other complex walking”, “running”, “stationary”, “unspecified arms activities while sitting/standing”, or “unspecified arms activities while walking”, with the class weights set to [50, 25, 10, 5, 5, 5] to primarily focus on walking. The second stage further classified windows in the “other complex walking” into one of the following: “walking with hands in pocket”, “walking with hands held stationary”, “walking with hand held next to the ear”, “walking with hand resting on the shoulder”, and “ walking with a briefcase/grocery bag”. The SVM model was configured with a Radial Basis Function (RBF) kernel, with a kernel coefficient, gamma, of 1/54.

Subsequently, we identified their longest continuous walking bouts by counting the greatest number of consecutive walking windows. To address brief pauses within extended walking bouts, gaps meeting the three criteria below were bridged: (1) not exceeding 60 seconds, (2) not exceeding both the following and preceding consecutive walking window durations and (3) not exceeding one-fifth of either the following or preceding consecutive walking window durations. The proportion of walks ≥ 8 seconds and 60 seconds were calculated through the cumulative exposure of walking durations. Steps were identified by cross-checking the peak signal detection with auto-correlation coefficients within each window identified as walking. A corrective factor derived from the unspecified arm activities was added to the total step count. We regrouped walks into episodes of eight steps each and calculated cadence by timing these episodes. We calculated the standard deviation of step times within each episode and derived step-time variability as the 95th percentile of the day. Step and stride regularity were derived as the normalized coefficients of the first and second peaks in the autocorrelation function and range between −1 to +1. Walking speed was estimated with SVM regression with ten fold validation (data partitioned at the participant level) by fitting (1) the participant’s sex, (2) the participant’s body height, (3) the median, (4) the interquartile range of the static-block-removed acceleration signal vector magnitude, (5) mean crude acceleration signal vector magnitude, (6) mean step time, and correlation coefficients between acceleration signals in the (7) x- and y-axes and (8) x- and z-axes in windows classified as walking with arm swings. The 95th percentile and median were identified as maximal and usual walking speeds. Sleep duration and bedtime were extracted from the wrist accelerometer data based on the van Hees approach [19].

### 2.2. Derivation of Normative Values for the Watch Walk Digital Biomarkers from the UK Biobank Dataset

The UK Biobank cohort comprised 503,317 participants. Sociodemographic characteristics, such as age, sex, ethnicity, and self-reported pace were captured from in-person baseline assessments from 2006 to 2010. From 2013 to 2015, 236,519 participants were randomly invited to wear wrist accelerometers (AX3 data loggers) for seven days on their dominant wrist to monitor physical activities, and 103,659 consented to participate [14]. Prior to the collection, all participants gave their written consent. The UKB study was approved by the National Information Governance Board for Health and Social Care and the North West Multicentre Research Ethics Committee.

Digital biomarkers that adhered to a normal distribution were summarized using means and standard deviations, and biomarkers exhibiting a highly skewed distribution were represented by medians and interquartile ranges. We presented descriptive statistics and normative values for participants achieving five complete days of wear time (*N* = 73,438) and those meeting the criterion of at least three dayswith 12 hours of wear time (*N* = 92,022) in two distinct columns. The distributions of digital biomarkers for the group with five complete days of wear time were visually illustrated through histograms.

Selected biomarkers represent their respective domains: steps per day (gait quantity), duration of the longest continuous walk (walk length distribution), maximal walking speed (gait speed and intensity), step regularity (gait quality), the proportion of walking with arm swings (walk hand positions), and sleep duration (Sleep). To evaluate sex differences in parametric measures, we used independent *t*-tests, and for non-parametric measures, Mann–Whitney U tests were applied. We also explored differences across age groups—45–54, 55–64, 65–74, and 75–79 years—using one-way analysis of variance (ANOVA) for parametric measures and Kruskal–Wallis tests for non-parametric measures. The visual representation of distribution differences was conducted with violin plots.

### 2.3. Reliability and External Validity of the Watch Walk Digital Biomarkers

For the development of these norms for the Watch Walk algorithms (v.2.3.0), we used Python (v 3.9.13) to enable widespread clinical and research use. An online platform that allows research groups to access these algorithms is currently under development and will be available at www.neura.edu.au/. The outputs from these analyses differ marginally from a previous implementation in MATLAB 9.7 (R2019b) [18].

Considering the differences between these programming environments, we performed a revalidation. The activity categorization algorithm accuracies were evaluated using 10-fold validation and presented in confusion matrices. Using data from the development study, the Watch Walk step time and walking speed were compared with electronic pathway measurements by mean absolute percentage error (MAPE). Further reliability and validation analyses were performed using the UK Biobank dataset. Test–retest reliability of the parameters was evaluated with intraclass correlation coefficients (2-way random effects, absolute agreement, mean of multiple measurements) for up to seven consecutive days. We compared maximal and usual walking speeds with self-reported walking pace categories: slow, average, and brisk. We further explored the relationship between step count, duration of the longest continuous walk, maximal walking speed, step regularity, and the proportion of walks with arm swings, and levels of self-reported mobility problems (ranging from severe to none). For these comparisons, one-way analysis of variance (ANOVA) or Kruskal–Wallis tests were used, depending on whether the distributions of digital gait biomarkers were parametric, and the results of these analyses were visually depicted using violin plots. Statistical analyses were performed using SAS Enterprise 8.3 software (8.3.0.103).

## 3. Results

### 3.1. Participant Characteristics

Of the 106,053 UK Biobank participants, 73,438 (69.2%) had at least five complete days of sensor wear time with valid data and 91,876 (86.8%) had at least three 12 hour days wear time. Table 1 compares the sociodemographic characteristics of the included and excluded participants. Significant, yet small (<1% in proportion), differences in age, sex, ethnicity, BMI status, and self-reported mobility problem status were evident.

### 3.2. Normative Values

Figure 1 and Appendix A show that sleep duration, bedtime, steps per day, walks ≥ 8 s, maximal and usual walking speed, cadence median and interquartile range, step regularity, and step and stride regularity were largely normally distributed. Longest continuous walking duration, walks ≥ 60 s and proportion of walks with various hand positions were skewed. Descriptive statistics of other digital sleep and gait biomarkers are presented in Table 2 and Appendix A. Appendix A report the 5th, 10th, 25th (lower quartile), 50th (median), 75th (upper quartile), 90th and 95th percentiles for each digital biomarker in each age group for men and women, respectively. The maximal and usual walking speed mean were 1.49 and 1.15 ms^−1^, respectively, with SD of 0.08 ms-^1^. The mean step count was 7749 (SD = 2739), and the mean step regularity was 65.4% (SD = 4.8%). The mean cadence was 101 (SD = 3.2) steps per minute. The median duration for the longest continuous walk duration was 333 seconds (interquartile range, IQR, 189 to 549).

Figure 2 displays the Watch Walk digital biomarkers profiles for four participants. Among them, one exhibited severe walking difficulties, one had slight walking difficulties and two reported no walking difficulties. These profiles showcase diverse performance combinations across various gait domains, highlighting the potential application of the reported normative values and the utility of identifying performance in different domains to individualise intervention.

### 3.3. Sex and Age Group Differences

Figure 3 and Figure 4 present the comparisons of sleep duration, steps per day, duration of the longest continuous walk, maximal walking speed, and step regularity and the proportion of walks with arm swings across sex and age groups, respectively. Women walked more and with more regular steps compared to men, but walked at a slower pace and the duration of their longest continuous walk was shorter. The mean sleep period times ± SD were 8.3 ± 1.0 hour for women and 8.5 ± 1.2 hour for men, t(73,436) = −26.53, *p* < 0.001. The mean step counts ± SD were 7979.9 ± 2747.6 for women and 7473.0 ± 2706.2 for men, t(73,436) = 25.09, *p* < 0.001. The median longest walk duration, and the corresponding interquartile range, were 330.4 [186.4–542.3] for women and 336.0 [192.0–556.7] for men (z = 656,559,841.5, *p* < 0.001). The mean durations of the longest walk ± SD were 407.0 ± 307.6 s for women and 424.9 ± 336.4 s for men, t(73,436) = −7.53, *p* < 0.001. The mean maximum walking speeds ± SD were 145.5 ± 6.8 cms^−1^ for women and 153.9 ± 7.6 cms^−1^ for men, t(73,436) = −156.97, *p* < 0.001. The mean step regularity ± SD was 66 ± 5% for women and 65 ± 5% for men, t(73,436) = 8.49, *p* < 0.001. The mean proportion of arm swing ± SD was 90 ± 8% for women and 92 ± 7% for men, t(73,436) = −33.89, *p* < 0.001.

Across four age groups (45–54, 55–64, 65–74, and 75–79), a consistent age-related decline was observed across various digital biomarkers. Mean sleep duration, with corresponding standard deviations, were 8.2 ± 1.0, 8.3 ± 1.0, 8.5 ± 1.1, and 8.5 ± 1.2 h, respectively (F_(3, 73,435)_ = 393.501, *p* < 0.001). The average step counts were 8200 ± 2689, 8024 ± 2769, 7349 ± 2669, and 6413 ± 2646, respectively (F_(3, 73,435)_ = 585.294, *p* < 0.001). Median durations of the longest walk in seconds and the corresponding interquartile ranges were 354 [209, 567], 355 [200, 574], 310 [174, 519], 247 [133, 449], respectively (H_(3, n=73,435)_ = 630.4, *p* < 0.001). Mean maximal walking speeds, with standard deviations, were 1.52 ± 0.08 ms^−1^, 1.50 ± 0.08 ms^−1^, 1.48 ± 0.08 ms^−1^, and 1.45 ± 0.08 ms^−1^, respectively (F_(3, 73,435)_ = 1353.187, *p* < 0.001). The average step regularity was 66 ± 4%, 66 ± 5%, 65 ± 5%, and 64 ± 5%, respectively (F_(3, 73,435)_ = 309.207, *p* < 0.001). Lastly, the average proportions of arm swing were 90 ± 8%, 90 ± 7%, 91 ± 7%, and 92 ± 7%, respectively (F_(3, 73,435)_ = 173.417, *p* < 0.001).

### 3.4. Reliability and External Validity of the Watch Walk Digital Biomarkers

Figure 5 and Figure 6 present the activity classification accuracy through confusion matrix tables. The walking activity class had a sensitivity of 94% and a precision of 93%. Within windows labelled as walking activity, the walking with arm swing subclass had a sensitivity of 94% and a specificity of 97%. Appendix A presents the mean absolute percentage error (MAPE) of sensor-based step time (ranging from 2.9 ± 4.3% to 5.2 ± 11.2%) and walking speed estimations (5.7 ± 8.5%). Figure 7 and Figure 8 illustrate strong agreement between maximal walking speed, usual walking speed, and steps per day with self-reported walking pace and the severity of walking problems, respectively. For individuals reporting slow, average, and brisk walking paces, the mean maximal walking speeds with corresponding standard deviations were 1.43 ± 0.08, 1.48 ± 0.08, and 1.52 ± 0.08 ms^−1^, respectively. Similarly, the mean usual walking speeds with corresponding standard deviations were 1.08 ± 0.07, 1.13 ± 0.08, and 1.16 ± 0.08 ms^−1^, respectively. Significant differences were observed between groups for both maximal walking speed (F_(2, 73,230)_ = 3052, *p* < 0.001) and usual walking speed (F_(2, 73,230)_ = 3014, *p* < 0.001). For individuals reporting severe, moderate, slight, and no mobility problems, significant differences were observed across various mobility metrics. The mean step counts, with corresponding standard deviations, were 5660 ± 2785, 6824 ± 2775, 7582 ± 2685, and 8035 ± 2665 for severe, moderate, slight, and no mobility problem groups, respectively (F_(3, 51,788)_ = 498.3, *p* < 0.001). The median durations of the longest walk in seconds, and the corresponding interquartile ranges, were 138 [65, 274], 233 [124, 423], 317 [179, 523], and 371 [218, 589], respectively, showing significant differences (H_(3, 51,792)_ = 1915.4, *p* < 0.001). Mean maximal walking speeds in centimetre per second, with standard deviations, were 144 ± 8, 145 ± 8, 148 ± 8, and 150 ± 8, also indicating significant variations (F_(3, 51,788)_ = 604.0, *p* < 0.001). The mean step regularity, with standard deviation, were 61 ± 6%, 63 ± 6%, 65 ± 5%, and 66 ± 4%, with significant differences (F(_3, 51,788_) = 146.8, *p* < 0.001). Lastly, the mean proportions of arm swing were 94 ± 6%, 93 ± 6%, 91 ± 7%, and 90 ± 7%, demonstrating significant group differences (F_(3, 51,788)_ = 275.604, *p* < 0.001).

## 4. Discussion

This study provides normative reference values for a range of digital gait characteristics—including walking speed, distribution, quality, and quantity—based on data from 92,022 and a subset of 73,438 UK individuals aged 45 to 79 years who wore a small wrist device for three and five days, respectively. Our findings revealed sex differences in walking behaviours; compared to men, women typically exhibited slower walking speeds and engaged in a smaller proportion of long walks yet walked more frequently and with greater regularity (indicating more stable gait). Additionally, we observed a gradual decline in walking performance with age, characterized by reductions in walking speed, shorter continuous walking durations, lower step counts and lower step regularity.

Considering the individuality and multitude of factors that influence daily-life walking, digital gait profiles derived from a wrist device may provide an unobtrusive new way to remotely identify risk areas for targeted intervention (Figure 2). For example, the faster (e.g., greater cadence), unstable (e.g., reduced stride regularity) and more intermittent (e.g., fewer walks ≥ 8 s) gait presented by a 63-year-old woman might be considered an early warning for their subsequent injurious fall. Conversely, the slower (e.g., lower cadence) but stable (e.g., increased stride regularity) gait presented by a 66-year-old woman may indicate a more careful approach to daily-life walking and lower risk of fall-related injuries.

The maximal walking speeds reported in this study (mean 1.49 ms^−1^) were comparable to those derived from a waist accelerometer in 254 community-dwelling Dutch adults (mean 1.49 ms^−1^) [7]. Conversely, faster usual walking speeds were observed for participants in the present study (mean 1.15 ms^−1^ vs. 0.90 ms^−1^), which may indicate environmental or cultural differences between cohorts. While maximal walking speed may be more indicative of an individual’s physical capacity, usual walking speed may be influenced by lifestyle and environmental factors. As the measurements were extracted from participants’ daily life, walking speeds may be impacted by terrain, weather conditions and urban design. Individuals living in cities with neighbourhoods more amenable to walking, such as higher street connectivity, more land use mix, and appropriate residential density, have higher step counts [20]. Future studies should investigate how city walkability and greenspace availability affect other daily-life walking measures.

The observed sex differences align with previous research, which has documented that women in both Germany [21] and Japan [22] walk slower but take more daily steps compared to men. These observations could reflect different physical capacities and behavioural patterns between men and women. Differences in walking speed may partly be attributed to women having lower muscle strength [23] and shorter stature [24] compared to men. This trend may also be partly attributed to the observation that more women than men walk for leisure [25], which generally has a slower pace. It is critical to examine these sex-specific patterns as they have direct implications for developing public health guidelines. An illustrative example is the female-focused physical activity policy introduced in Japan [26,27], which aimed to reduce sex disparities in physical activity. These interventions resulted in greater improvements in walking speed and physical activity levels for women compared to men [26,28].

Using a conservative wear time criterion of at least five complete days of valid data, approximately 69% of individuals who consented to participate were included in the analysis. This proportion is comparable to, or even higher than, previous studies, indicating a higher level of acceptability. Prior research on physical activity estimation has suggested that a more lenient criterion can provide reliable estimates. For instance, Airlie et al. [29] recommended at least eight hours on four days to reliably estimate physical activity in older care home residents. Buckley et al. [30] found that two full days were needed to achieve an intraclass correlation coefficient (ICC) of 0.8 for step count measurement in residential-aged care settings. Czhech et al. [31] posited that two to three days of monitoring are sufficient to estimate both usual and maximal walking speed in healthy volunteers. Furthermore, Mueller et al. [32] proposed a criterion of at least three hours on any three days to estimate the step count in frail older individuals reliably. Such criteria are more practical and feasible and would yield sufficiently reliable estimates for clinical research and practice. By adhering to the criterion of at least three days of 12 hours of weartime, digital gait biomarkers could be extracted from 87% of recruited participants.

We acknowledge limitations that relate to the generalizability of the findings. First, the participation rate was higher among women, older age-groups and more affluent individuals, which may indicate the study participants were more health conscious than the general population. This “healthy volunteer” selection bias is reflected by the sample having lower all-cause mortality and cancer incidence than the age-matched general population [33]. Second, as all participants were residents of the United Kingdom, the normative values may not generalize to all other countries.

## 5. Conclusions

This study provides large-scale normative data for a range of wrist-sensor-derived digital sleep and gait biomarkers from real-world data. It provides benchmarks for identifying deviations from the norm for use in clinical practice and future research applications.

## Figures and Tables

**Figure 1 sensors-24-05159-f001:**
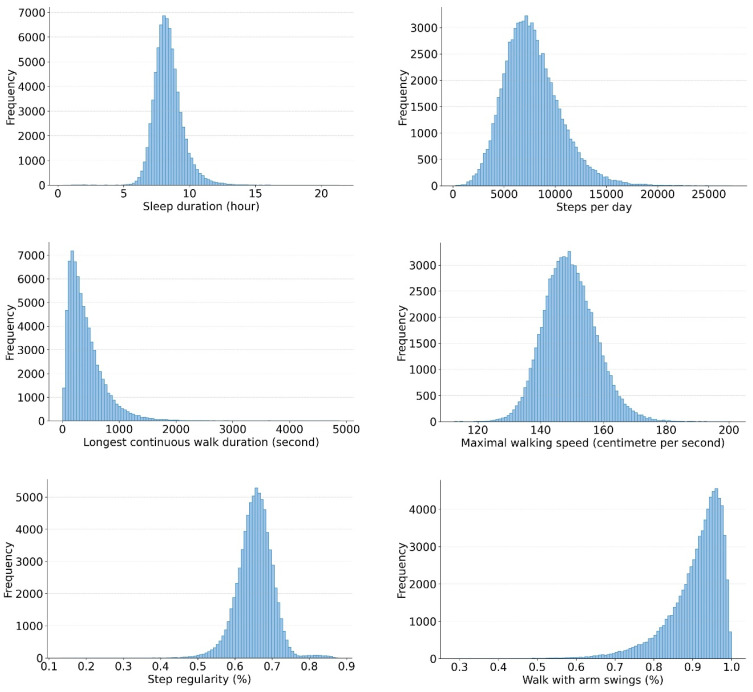
Histograms illustrating the distribution of selected sleep, gait quantity, distribution, speed quality, and hand position biomarkers.

**Figure 2 sensors-24-05159-f002:**
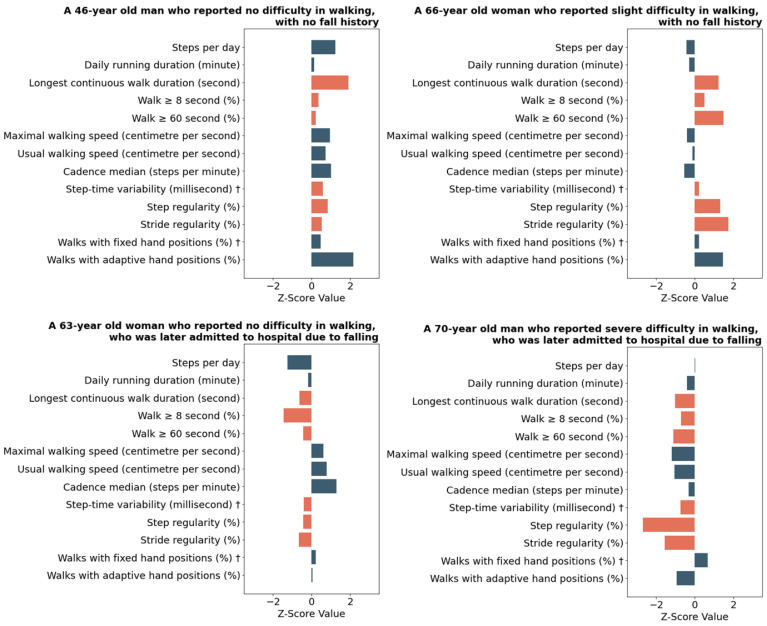
Digital Gait Biomarker (DGB) z-score output for four participants. Test scores represent standardised (z) scores, reference to the total sample (*n* = 73,438), to allow direct comparison in performance between participants. Each unit signifies one standard deviation. A score of zero reflects average performance relative to the study population, whereas positive and negative scores indicate performances that are above and below average, respectively. † indicates the direction of the digital gait biomarker is reversed, so that a positive score always indicates a better performance. Colour differences distinguish biomarker domains.

**Figure 3 sensors-24-05159-f003:**
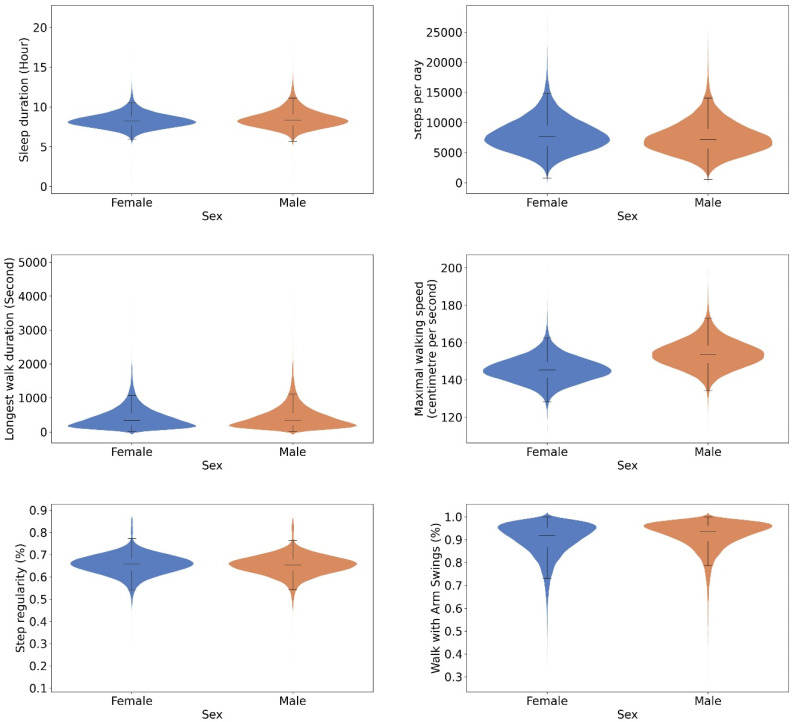
Violin plots illustrating the comparison of selected digital gait biomarkers between male and female. Widths of the violine plots, kernel densities; top and bottom of the violine plots, 1st and 99th percentiles; top and bottom of the box, upper and lower quartiles; line inside of the box, median; and caps of the whisker, the highest and lowest values within 1.5 times the IQR from the upper and lower quartiles, respectively. Significant sex differences were observed across various digital biomarkers.

**Figure 4 sensors-24-05159-f004:**
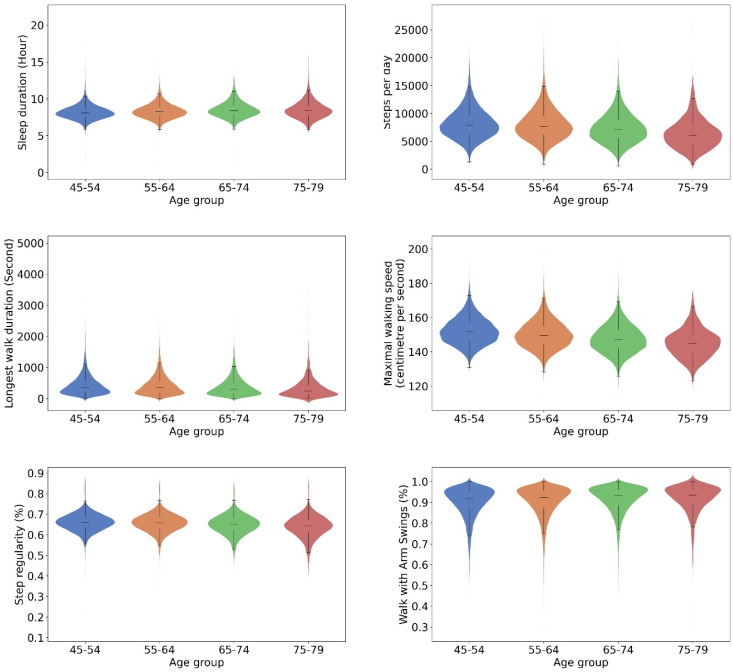
Violin plots illustrating the comparison of selected gait biomarkers between age groups. Widths of the violine plots, kernel densities; top and bottom of the violine plots, 1st and 99th percentiles; top and bottom of the box, upper and lower quartiles; line inside of the box, median; and caps of the whisker, the highest and lowest values within 1.5 times the IQR from the upper and lower quartiles, respectively.

**Figure 5 sensors-24-05159-f005:**
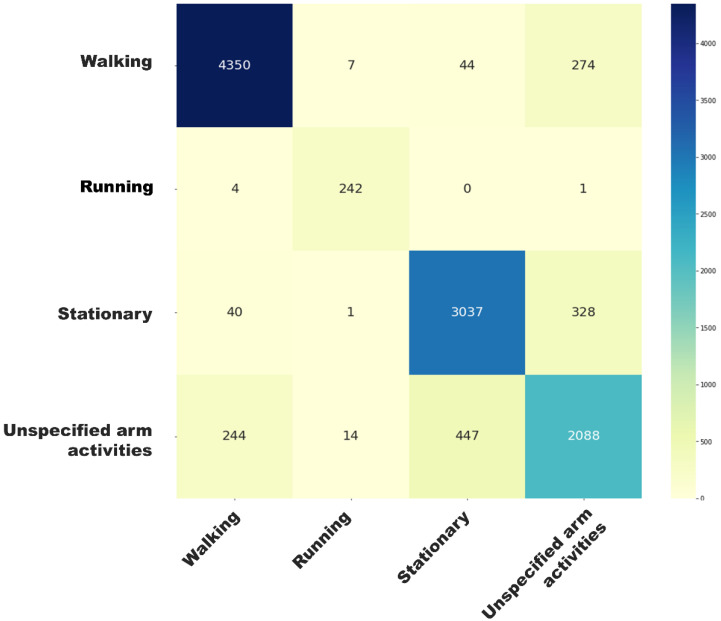
Confusion matrix of stage 1 classification.

**Figure 6 sensors-24-05159-f006:**
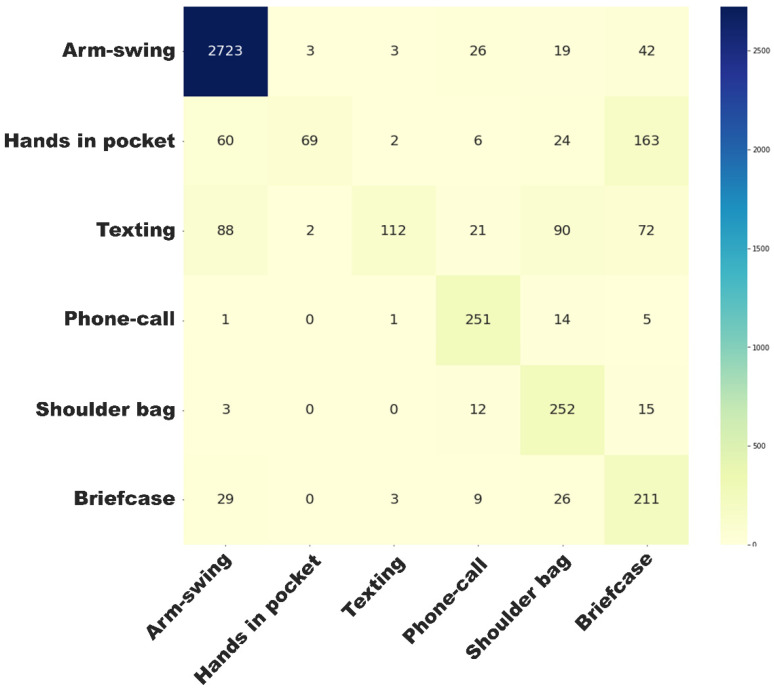
Confusion matrix of stage 2 classification.

**Figure 7 sensors-24-05159-f007:**
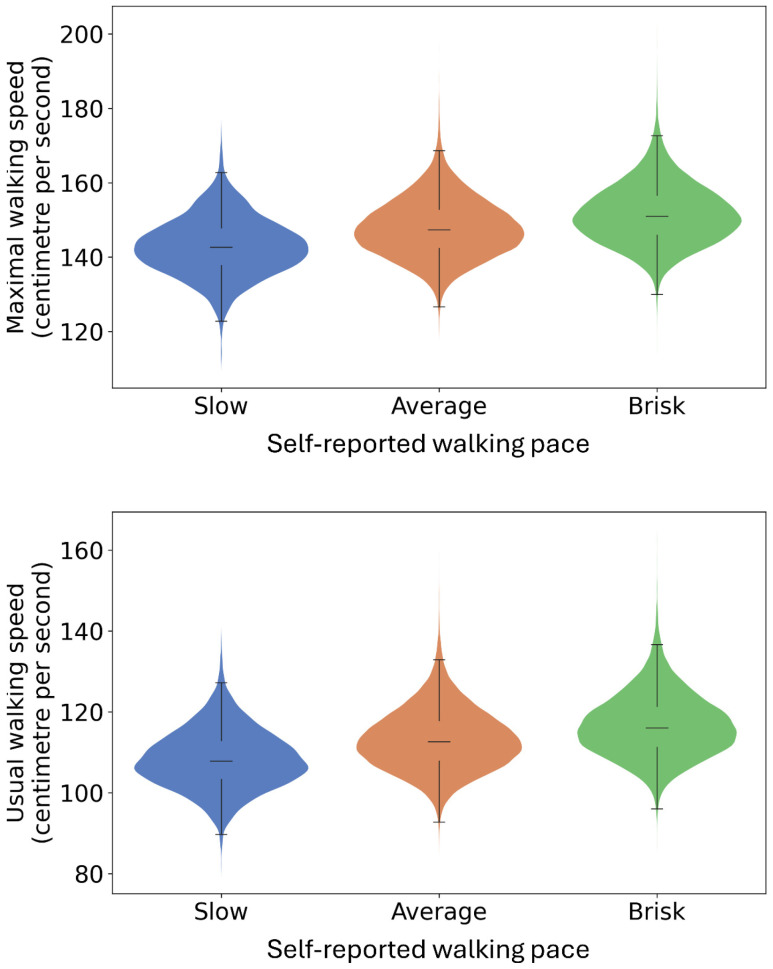
Maximal and usual walking speed for people who reported slow (*n* = 3431), steady (*n* = 35,319) and brisk (*n* = 34,483) walking paces. *N* = 73,233. Widths of the violine plots, kernel densities; top and bottom of the violine plots, 1st and 99th percentiles; top and bottom of the box, upper and lower quartiles; line inside of the box, median; and caps of the whisker, the highest and lowest values within 1.5 times the IQR from the upper and lower quartiles, respectively.

**Figure 8 sensors-24-05159-f008:**
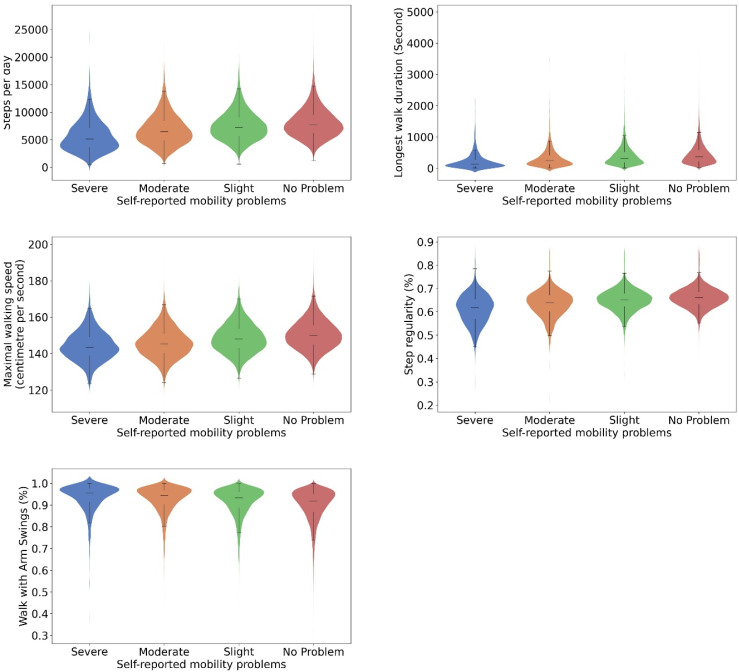
Digital gait biomarkers for people who reported severe (*n* = 1053), moderate (*n* = 3572) and slight (*n* = 9533) and no problems (*n* = 37,634) with walking. *N* = 51,792. Widths of the violine plots, kernel densities; top and bottom of the violine plots, 1st and 99th percentiles; top and bottom of the box, upper and lower quartiles; line inside of the box, median; and caps of the whisker, the highest and lowest values within 1.5 times the IQR from the upper and lower quartiles, respectively.

**Table 1 sensors-24-05159-t001:** Participant characteristics.

*n* (%) Unless Otherwise Stated	Excluded (*N* = 11,785)	Included 5 Days (*N* = 73,438)	Included 3 Days (*N* = 91,876)	Total (*N* = 103,661)	*p*-Value between Excluded and Included 5 Days
Age Mean (SD)	61.7 (7.9)	62.4 (7.8)	62.3 (7.9)	62.2 (7.9)	<0.001 †
Female	6501 (55.2%)	39,636 (54.0%)	51,766 (56.3%)	58,267 (56.2%)	<0.001
Ethnicity					<0.001
White	11,252 (95.9%)	70,967 (97.0%)	88,727 (96.9%)	99,979 (96.8%)	
Black	132 (1.1%)	733 (1.0%)	948 (1.0%)	1080 (1.0%)	
Asian	250 (2.1%)	954 (1.3%)	1198 (1.3%)	1448 (1.4%)	
Other	94 (0.8%)	541 (0.7%)	693 (0.8%)	787 (0.8%)	
Prefer not to answer	57	243	310	367	
BMI status					<0.001
Normal	4353 (36.9%)	28,275 (38.5%)	35,655 (38.8%)	40,008 (38.6%)	
Underweight	239 (2.0%)	445 (0.6%)	587 (0.6%)	826 (0.8%)	
Overweight	4815 (40.9%)	30,387 (41.4%)	37,738 (41.1%)	42,553 (41.1%)	
Obese	2378 (20.2%)	14,331 (19.5%)	17,896 (19.5%)	20,274 (19.6%)	
Self-reported mobility problem					<0.01
Unable to walk	30 (0.4%)	44 (0.1%)	62 (0.1%)	92 (0.1%)	
Severe	194 (2.5%)	1047 (2.0%)	1306 (2.0%)	1500 (2.1%)	
Moderate	571 (7.4%)	3561 (6.9%)	4518 (7.0%)	5089 (7.0%)	
Slight	1444 (18.6%)	9525 (18.4%)	11,935 (18.5%)	13,379 (18.5%)	
No problem	5526 (71.2%)	37,607 (72.6%)	46,688 (72.4%)	52,214 (72.2%)	
Prefer not to answer	4020	21,654	27,367	31,387	
With history of major depression	233 (2.0%)	1150 (1.6%)	1492 (1.6%)	1725 (1.7%)	<0.001
With Type 2 Diabetes Mellitus history	325 (2.8%)	2023 (2.8%)	2479 (2.7%)	2804 (2.7%)	0.12
With stroke history	157 (1.3%)	904 (1.2%)	1176 (1.3%)	1333 (1.3%)	0.01
With Parkinson’s disease history	27 (0.2%)	121(0.2%)	148 (0.2%)	175 (0.2%)	0.61
With dementia history	9 (0.1%)	35 (0.0%)	52 (0.1%)	61 (0.1%)	0.02

Chi-square test *p*-value unless specified; †, independent *t*-test *p*-value.

**Table 2 sensors-24-05159-t002:** The mean and standard deviation, or median and interquartile range of digital sleep and gait biomarkers, based on a minimum wear time of five complete days.

	Sex		Age Group					ICC
Digital Sleep and Gait Biomarkers, Units	Female (*N* = 39,636)	Male (*N* = 33,802)	45 to 54 (*N* = 14,620)	55 to 64 (*N* = 25,112)	65 to 74 (*N* = 31,361)	75 to 79 (*N* = 2345)	Total (*N* = 73,438)	
**Sleep**								
Sleep duration, hour †	8.28 (0.96)	8.49 (1.18)	8.17 (0.97)	8.32 (1.03)	8.51 (1.12)	8.54 (1.17)	8.38 (1.07)	0.92 [0.85, 0.96]
Bedtime, hour of the day †	23:17 (1 hr 4 min)	23:11 (1 hr 13 min)	23:13 (1 hr 11 min)	23.14 (1 h 9 min)	23:16 (1 hr 7min)	23:15 (1 hr 7 min)	23:15 (1 hr 8 min)	0.91 [0.84, 0.96]
**Gait Quantity**								
Steps per day †	7981 (2746)	7477 (2705)	8208 (2689)	8041 (2765)	7396 (2675)	6487 (2622)	7749 (2739)	0.96 [0.92, 0.98]
**Walk Length distribution**								
Longest continuous walk duration, second ‡	330.7 [186.4, 542.4]	336.0 [192.0, 556.8]	354.4 [210.7, 566.3]	355.3 [200.0, 575.4]	313.3 [176.6, 524.0]	252.6 [136.6, 450.9]	332.8 [188.8, 548.7]	0.93 [0.88, 0.97]
% Walk ≥ 8 s †	37.8 (5.20)	40.3 (5.78)	40.4 (5.20)	39.5 (5.47)	38.0 (5.67)	36.4 (6.09)	38.9 (5.63)	0.96 [0.94, 0.98]
% Walk ≥ 60 s ‡	1.5 [1.0, 2.6]	1.9 [1.0, 3.3]	1.9 [1.0, 3.1]	1.8 [1.0, 3.0]	1.5 [0.8, 2.7]	1.3 [5.8, 2.5]	1.7 [8.5, 2.9]	0.92 [0.86, 0.97]
**Gait Speed and Intensity**								
Maximal walking speed, centimetre per second †	145.5 (6.78)	153.9 (7.61)	152.3 (7.87)	150.1 (8.03)	147.7 (8.18)	145.3 (8.30)	149.4 (8.29)	0.98 [0.97, 0.99]
Usual walking speed, centimetre per second †	111.2 (6.64)	118.5 (7.65)	117.7 (7.59)	115.4 (7.69)	112.8 (7.80)	110.5 (7.77)	114.6 (7.98)	0.97 [0.95, 0.99]
Cadence median, spm †	101.6 (3.17)	100.1 (3.07)	101.1 (3.12)	101.0 (3.18)	100.7 (3.25)	100.6 (3.51)	100.9 (3.22)	0.89 [0.81, 0.95]
**Gait Quality**								
Step-time variability, millisecond †	57.7 (12.3)	64.7 (15.2)	59.3 (13.1)	60.2 (14.1)	62.1 (14.5)	63.7 (15.1)	60.9 (14.2)	0.95 [0.91, 0.98]
Step regularity, % †	65.6 (4.8)	65.3 (4.8)	66.1 (4.4)	65.8 (4.6)	64.9 (5.0)	64.0 (5.3)	65.4 (4.8)	0.94 [0.89, 0.97]
Stride regularity, % †	52.3 (4.7)	52.7 (4.5)	52.9 (4.4)	52.7 (4.5)	52.2 (4.7)	51.6 (4.8)	52.5 (4.6)	0.94 [0.89, 0.97]
**Walk hand positions**								
% Walk with Arm Swings‡	91.8 [86.5, 95.4]	93.5 [89.2, 96.2]	91.8 [86.7, 95.2]	92.4 [87.2, 95.6]	93.2 [88.4, 96.2]	93.6 [89.3, 96.5]	92.6 [87.7, 95.8]	0.90 [0.82, 0.96]
% Texting‡	0.5 [0.3, 0.8]	0.4 [0.2, 0.7]	0.4 [0.3, 0.7]	0.4 [0.2, 0.7]	0.4 [0.2, 0.8]	0.5 [0.3, 0.9]	0.4 [0.2, 0.8]	0.90 [0.81, 0.95]
% Phone call‡	0.4 [0.1, 0.9]	0.3 [0.1, 0.8]	0.4 [0.1, 0.8]	0.4 [0.1, 0.9]	0.3 [0.1, 0.8]	0.4 [0.1, 0.9]	0.4 [0.1, 0.8]	0.80 [0.65, 0.91]
**%** Hands in pockets ‡	1.0 [0.5, 1.7]	0.8 [0.5, 1.5]	1.2 [0.7, 2.1]	1.0 [0.5, 1.7]	0.8 [0.4, 1.3]	0.6 [0.4, 1.1]	0.9 [0.5, 1.6]	0.91 [0.84, 0.96]
% Shoulder bag ‡	3.6 [1.8, 6.6]	2.9 [1.5, 5.5]	3.7 [1.9, 6.6]	3.4 [1.7, 6.3]	3.0 [1.5, 5.7]	2.8 [1.3, 5.3]	3.3 [1.6, 6.1]	0.89 [0.79, 0.95]
% Briefcase ‡	1.6 [0.8, 3.0]	0.9 [0.5, 1.7]	1.3 [0.7, 2.6]	1.2 [0.6, 2.5]	1.1 [0.5, 2.3]	1.0 [0.5, 2.1]	1.2 [0.6, 2.4]	0.93 [0.87, 0.97]
% Static which includes texting and cell phone ‡	1.0 [0.5, 1.8]	0.8 [0.4, 1.5]	0.9 [0.5, 1.6]	0.9 [0.5, 1.6]	0.9 [0.5, 1.6]	1.0 [0.5, 1.8]	0.9 [0.5, 1.6]	0.87 [0.77, 0.94]
% Adaptive which includes shoulder bag, brief case and hands in pockets ‡	6.9 [3.7, 11.7]	5.4 [3.0, 9.2]	7.0 [3.9, 11.8]	6.4 [3.6, 11.0]	5.6 [3.1, 9.8]	5.1 [2.7, 8.8]	6.1 [3.4, 10.6]	0.91 [0.83, 0.96]

Bold text indicates biomarker domains. All *p*-value < 0.0001, except *p*-value for longest continuous walk duration between sex = 0.02. † indicates that the variable is presented as the mean (standard deviation). ‡ indicates that the variable is presented as the median [interquartile range]. Hr, hour; Min, minute.

## Data Availability

Restrictions apply to the availability of these data. Data were obtained from UK Biobank (accessed on 11 September 2023) and are available at https://www.ukbiobank.ac.uk/ with the permission of UK Biobank.

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
