# Peer review of "Daily-Life Walking Speed, Quality and Quantity Derived from a Wrist Motion Sensor: Large-Scale Normative Data for Middle-Aged and Older Adults"

_sensors, 2024, doi:10.3390/s24165159_

Round 1

Reviewer 1 Report

Comments and Suggestions for Authors

Issues:

1. In the paper, it is mentioned: "We recruited participants aged 19 to 91". How many are men or women? Are they all healthy?

2. Which are the parameters from the bandpass filters used in "accelerometric signals and filtered it with Butterworth bandpass filters"? How many filters were used? Generalizing this observation generated by these questions: the presentation of the conditions in which the analyses are made is superficial. This superficiality does not allow for restoration of the way of obtaining the results and their independent verification. Another example the authors mention the use of an SVM-type neural network – do these neural networks have a linear kernel or a non-linear one? Is it the kernel polynomial or RBF? What is the size of the kernel? And so on ...

3. The link from the "Supplementary Materials" is broken.

4. Table 1 presents the participants' characteristics in data acquisition and analysis. The table shows the level of education as a characteristic of the participants - a characteristic totally unimportant for the present study. Instead, the state of health of the locomotor system or the presence of some neuronal disorders are not considered.

5. Use the power when defining this speed in: "usual walking speed mean were 1.49 and 169 1.15 ms-1".   ... or use the slash [m/s]. The same is true here: "The maximal walking speeds reported in this study (mean 1.49 ms-1)" and in the paper's extent.

Comments on the Quality of English Language

Minor modification.

Author Response

Comment 1: In the paper, it is mentioned: "We recruited participants aged 19 to 91". How many are men or women? Are they all healthy?

Response 1: Thank you for pointing this out. We have added the proportion of women and their health status to better characterise the participants:

We recruited 101 participants aged 19 to 91 (mean 47, SD:18) (67% female) from Sydney, Australia and Hong Kong, China. All participants reported no osteoporosis, neurological impairment, recent fractures, recent joint replacements or existing conditions that restrict exercise tolerance.

Comment 2: Which are the parameters from the bandpass filters used in "accelerometric signals and filtered it with Butterworth bandpass filters"? How many filters were used? Generalizing this observation generated by these questions: the presentation of the conditions in which the analyses are made is superficial. This superficiality does not allow for restoration of the way of obtaining the results and their independent verification. Another example the authors mention the use of an SVM-type neural network – do these neural networks have a linear kernel or a non-linear one? Is it the kernel polynomial or RBF? What is the size of the kernel? And so on ...

Response 2: We have written a more comprehensive description of the bandpass filter, the extracted features and the SVM model to facilitate reproducibility:

We derived the sample level Euclidean norm from the annotated accelerometric signals and filtered it with a Butterworth bandpass filter (5th order, 0.25 Hz low cutoff frequency, 2.5 Hz high cutoff frequency, and 100 Hz sampling frequency). The filtered signals were then segmented into non-overlapping 4-second windows. Fifty-four features were extracted for training and validating a two-stage multiclass support vector machine (SVM) classification model with ten k-fold validation at the participant level. In the first stage, the 4-sedcond window was first classified into “walks with arm-swing”, “other complex walking”, “running”, “stationary”, “unspecified arms activities while sitting/ standing”, or “unspecified arms activities while walking”, with the class weights set to [50, 25, 10, 5, 5, 5] to primarily focus on walking. The second stage further classified windows in the “other complex walking” into one of the following: “walking with hands in pocket”, “walking with hands held stationary”, “walking with hand held next to the ear”, “walking with hand resting on the shoulder”, and “ walking with a briefcase/ grocery bag”. The SVM model was configured with a Radial Basis Function (RBF) kernel, with a kernel coefficient, gamma, of 1/54.

Comment 3: The link from the "Supplementary Materials" is broken.

 Response 3: Please find the supplementary materials located at the end of the document.

Comment 4:  Table 1 presents the participants' characteristics in data acquisition and analysis. The table shows the level of education as a characteristic of the participants - a characteristic totally unimportant for the present study. Instead, the state of health of the locomotor system or the presence of some neuronal disorders are not considered.

Response 4: We have removed the less relevant characteristic “level of education” and added the two characteristics “self-reported mobility” and “history of Parkinson’s disease” in Table 1.

Comment 5:  Use the power when defining this speed in: "usual walking speed mean were 1.49 and 1.15 ms-1".   ... or use the slash [m/s]. The same is true here: "The maximal walking speeds reported in this study (mean 1.49 ms-1)" and in the paper's extent.

Response 5: Thank you for pointing this out. We have superscripted “-1” to indicate that it is a power throughout the paper.

Reviewer 2 Report

Comments and Suggestions for Authors

Dear Authors,

This study leverages wrist-worn sensor data from UK Biobank participants to establish normative daily-life walking data, stratified by age and sex, to provide benchmarks for research and clinical practice. One-week wrist sensor data from 106,053 UK Biobank participants were obtained using the Watch Walk method. 19 daily-life walk and sleep-related biomarkers, including walking speed, step regularity, duration of longest continuous walk, step count, bedtime and sleep duration were extracted. This study provides large-scale normative data and benchmarks for wrist-sensor-derived digital gait and sleep biomarkers from real-world data for future research and clinical applications. It was shown that walking speed, step count, longest walk duration, and step regularity decreased with age.

The results obtained are in good agreement with other similar studies. Currently, the problem of assessing behavior in the wild is widely studied and covered in many similar publications.

To increase reader interest, it is necessary to improve the readability of the manuscript.

The purpose of the study and its novelty should be presented more clearly.

The study design should be explained more clearly and logically. 1) Study groups must be characterized; 2) the research points and conditions for measuring body functions must be explained; 3) all devices used should be noted; 4) signal processing and parameters must be presented; 5) a statistical analysis plan must be presented.

The results should be presented more clearly. Parameters must be presented uniformly in tables. Use one of the statistical parameters: median and interquartile range, as a less powerful indicator is preferable.

Figures 3 and 4 are provided with lengthy comments. This information is difficult to compare with figures.

The main result regarding age characteristics is unclear from the maintext and figures.

My overall comment, the manuscript is not ready for publication in its present form, as it has low readability and needs to be edited.

Comments on the Quality of English Language

several grammatical errors, typos should be corrected

Author Response

Comment 1: This study leverages wrist-worn sensor data from UK Biobank participants to establish normative daily-life walking data, stratified by age and sex, to provide benchmarks for research and clinical practice. One-week wrist sensor data from 106,053 UK Biobank participants were obtained using the Watch Walk method. 19 daily-life walk and sleep-related biomarkers, including walking speed, step regularity, duration of longest continuous walk, step count, bedtime and sleep duration were extracted. This study provides large-scale normative data and benchmarks for wrist-sensor-derived digital gait and sleep biomarkers from real-world data for future research and clinical applications. It was shown that walking speed, step count, longest walk duration, and step regularity decreased with age.

The results obtained are in good agreement with other similar studies. Currently, the problem of assessing behavior in the wild is widely studied and covered in many similar publications.

To increase reader interest, it is necessary to improve the readability of the manuscript.

Response 1: We would like to thank reviewer 2 for their comments. Please find our point-to-point response below.

Comment 2: The purpose of the study and its novelty should be presented more clearly.

Response 2: In response to Comment 2, we have revised the introduction section to more clearly present the purpose and novelty of our study. Specifically, in paragraph 3, we elaborate on the lack of normative data on daily-life walking derived from wrist-worn sensors. Despite the inclusion of wrist-worn accelerometers in recent large-scale population studies like the UK Biobank, the National Health and Nutritional Examination Survey (NHANES), the Rotterdam Study, and the Irish Longitudinal Study on Ageing (TILDA), there are no publications on the normative value of wrist-sensor-based daily life gait characteristics. Additionally, we have rephrased the introduction section to better highlight the study's purpose.

Wrist-worn accelerometers have now been used in recent large-scale population studies, including the UK Biobank [14], National Health and Nutritional Examination Survey (NHANES) [15], the Rotterdam study [16], and the Irish Longitudinal Study on Ageing (TILDA) [17]. However, as of now, normative values for daily-life walking performance based on wrist-worn sensors have not been derived.

To address this gap, we analyzed one-week wrist sensor data from 106,053 UK Biobank participants using the Watch Walk method [18]. We extracted the normative values of 19 daily-life walk- and sleep-related biomarkers, including walking speed, step regularity, duration of longest continuous walk, step count, bedtime and sleep duration. We assessed the proportion of participants included in the analysis relative to two different wear-time, three- and five-day thresholds, derived normative values for walking parameters for the cohort, and compared these measures between men and women and across different age groups.

Comment 3: The study design should be explained more clearly and logically. 1) Study groups must be characterized; 

Response 3: We have expanded Table 1. Participant characteristics by adding “self-reported mobility problems” and “History of  Parkinson’s disease” to better characterise the study group

Comment 4: 2) the research points and conditions for measuring body functions must be explained; 

Response 4: We have restructured the text to provide a clearer explanation of the sensor positions and the conditions under which the walking data was captured, as detailed below:

An accelerometer (AX3 data logger, Axivity Ltd, UK) was positioned over the participant’s dominant wrist and were videotaped during the data collection. They conducted thirty minutes of daily activities and walked on an electronic walkway (GAITRite, CIR System Inc, USA) with six different hand positions, including walking (1) with arm-swing, (2) with hands in pockets, (3) with hands held static in front of the body (texting), (4) with the dominant handheld next to the ear (phonecall), (5) while carrying a bag over the shoulder, and (6) while carrying a briefcase/grocery bag at slow, usual and fast paces.

Comment 5:  3) all devices used should be noted;

Response 5: The devices used are reported in sections 2.1 and 2.2:

An accelerometer (AX3 data logger, Axivity Ltd, UK) was positioned over the participant’s dominant wrist and were videotaped during the data collection. They conducted thirty minutes of daily activities and walked on an electronic walkway (GAITRite, CIR System Inc, USA) with six different hand positions, including walking (1) with arm-swing, (2) with hands in pockets, (3) with hands held static in front of the body (texting), (4) with the dominant hand held next to the ear (phonecall), (5) while carrying a bag over the shoulder, and (6) while carrying a briefcase/grocery bag at slow, usual and fast paces.

From 2013 to 2015, 236,519 participants were randomly invited to wear wrist accelerometers (AX3 data loggers) for seven days on their dominant wrist to monitor physical activities.

Comment 6:  4) signal processing and parameters must be presented

Response 6: We have further elaborated on the signal processing techniques used and the machine learning parameters as below:

We referenced the video recordings for the synchronization and annotation of the accelerometric signals and ground-truth electronic walkway measurements, including walking speed and step time. We derived the sample level Euclidean norm from the annotated accelerometric signals and filtered it with a Butterworth bandpass filter (5th order, 0.25 Hz low cutoff frequency, 2.5 Hz high cutoff frequency, and 100 Hz sampling frequency). The filtered signals were then segmented into non-overlapping 4-second windows. Fifty-four features were extracted for training and validating a two-stage multiclass support vector machine (SVM) classification model with ten k-fold validation at the participant level. In the first stage, the 4-second window was first classified into “walks with arm-swing”, “other complex walking”, “running”, “stationary”, “unspecified arms activities while sitting/ standing”, or “unspecified arms activities while walking”, with the class weights set to [50, 25, 10, 5, 5, 5] to focus on walking. The second stage further classifies windows in the “other complex walking” into one of the following: “walking with hands in pocket”, “walking with hands held stationary”, “walking with hand held next to the ear”, “walking with hand resting on the shoulder”, and “ walking with a briefcase/ grocery bag”. The SVM model was configured with a Radial Basis Function (RBF) kernel, with a kernel coefficient, gamma, of 1/54.

Comment 7:  a statistical analysis plan must be presented

Response 7: A separate statistical analysis plan was not completed for this study. The sttatistical analyses undertaken are presented in section 2.3:

Considering the differences between these programming environments, we performed a revalidation (see supplementary material). The activity categorization algorithm accuracies were evaluated using 10-fold validation and presented in confusion matrices. Using data from the development study, the Watch Walk step time and walking speed were compared with electronic pathway measurements by mean absolute percentage error (MAPE). Further reliability and validation analyses were performed using the UK Biobank dataset. Test-retest reliability of the parameters was evaluated with intraclass correlation coefficients (2-way random effects, absolute agreement, mean of multiple measurements) for up to seven consecutive days. We compared maximal and usual walking speeds with self-reported walking pace categories: slow, average, and brisk. We further explored the relationship between step count, duration of the longest continuous walk, maximal walking speed, step regularity, and the proportion of walks with arm swings, and levels of self-reported mobility problems (ranging from severe to none). For these comparisons, one-way analysis of variance (ANOVA) or Kruskal-Wallis tests were used, depending on whether the distributions of digital gait biomarkers were parametric, and the results of these analyses were visually depicted using violin plots. Statistical analyses were performed using SAS Enterprise 8.3 software.

Comment 8:  The results should be presented more clearly. Parameters must be presented uniformly in tables. Use one of the statistical parameters: median and interquartile range, as a less powerful indicator is preferable.

Response 8: We have carefully considered reviewer 2’s recommendation and appreciate the importance of presenting the data uniformly. In Table 2, we have used both the mean and standard deviation or the median and interquartile range, depending on whether the measure is parametric or not. This approach allows us to accurately represent the data’s distribution and variability according to their statistical characteristics.

Comment 9:  Figures 3 and 4 are provided with lengthy comments. This information is difficult to compare with figures.

Response 9: We have carefully considered Reviewer 2’s recommendation and recognise the importance of maintaining readability in figure captions. However, we feel the detailed captions for Figures 3 and 4 are irequired as they provide the key statistics illustrated in the violin plots, and that  this comprehensive information enhances the reader's understanding and comparison with the figures. Therefore, we prefer to retain the current format to maintain clarity and thoroughness.

Comment 10:  The main result regarding age characteristics is unclear from the maintext and figures.

Response 10: We have modified the text in section 3.3 to clarify the association between age and the digital gait biomarkers:

Comparing the four age groups ( 45-54, 55-64, 65-74, 75-79 years old), a consistent age-related decline was found in step count, maximal and usual walking speeds, step regularity, and the duration of the longest continuous walk ((F(3, 73230) = 130 to 1353, all p-values < 0.01). “

Reviewer 3 Report

Comments and Suggestions for Authors

This paper presents a significant contribution to the field of gait analysis and digital health by establishing normative daily-life walking data using wrist-worn sensors from a large cohort of UK Biobank participants. The study's rigorous approach and robust dataset offer valuable benchmarks for both research and clinical practice.

Given its comprehensive dataset, methodological rigor, and practical applications,  I recommend publishing this paper.

Author Response

Comment 1: This paper presents a significant contribution to the field of gait analysis and digital health by establishing normative daily-life walking data using wrist-worn sensors from a large cohort of UK Biobank participants. The study's rigorous approach and robust dataset offer valuable benchmarks for both research and clinical practice.

Given its comprehensive dataset, methodological rigor, and practical applications,  I recommend publishing this paper.

Response 1: We would like to thank reviewer 3 for their review and positive feedback.

Reviewer 4 Report

Comments and Suggestions for Authors

This manuscript presents a study provides normative data for a range of wrist-sensor-derived digital sleep and gait biomarkers in middle-aged and older adults. The data were collected through a wrist-worn sensor and processed using a method previously developed by the authors (Watch Walk method).

In Chan et al. (doi.org/10.1038/s41598-022-20327-z), the authors have described the method development and implementation that allows for the retrieval of digital gait biomarkers from daily life activity. The method was validated through its implementation in the UK Biobank dataset. The same dataset is used in the present study.

The test-retest reliability showed excellent results for this implementation. Therefore, the data provided by this work could serve as a reference for clinical practice and research. Moreover, they could be useful in identifying individuals who deviate from the normalized data, whether due to pathological or specific physical conditions. However, as the authors clearly exposed, the limitations of these results in relation to the sociodemographic characteristics of the participants must be considered.

To enhance the manuscript quality and to improve the study comprehension, I suggest adding a brief explanation about the Watch Walk method and its implementation. Although the authors' previous work (Chan et al, 2018) is cited at the introduction section, it is important that the present manuscript includes a brief description of the method and its scope both at the introduction and the methods section.

Author Response

Comment 1: This manuscript presents a study provides normative data for a range of wrist-sensor-derived digital sleep and gait biomarkers in middle-aged and older adults. The data were collected through a wrist-worn sensor and processed using a method previously developed by the authors (Watch Walk method).

In Chan et al. (doi.org/10.1038/s41598-022-20327-z), the authors have described the method development and implementation that allows for the retrieval of digital gait biomarkers from daily life activity. The method was validated through its implementation in the UK Biobank dataset. The same dataset is used in the present study.

The test-retest reliability showed excellent results for this implementation. Therefore, the data provided by this work could serve as a reference for clinical practice and research. Moreover, they could be useful in identifying individuals who deviate from the normalized data, whether due to pathological or specific physical conditions. However, as the authors clearly exposed, the limitations of these results in relation to the sociodemographic characteristics of the participants must be considered.

To enhance the manuscript quality and to improve the study comprehension, I suggest adding a brief explanation about the Watch Walk method and its implementation. Although the authors' previous work (Chan et al, 2018) is cited at the introduction section, it is important that the present manuscript includes a brief description of the method and its scope both at the introduction and the methods section.

Response 1: Thank you for this very constructive comment. We have rewritten a paragraph in the Introduction section to briefly describe the WatchWalk method:

To address this gap, we analyzed one-week wrist sensor data from 106,053 UK Biobank participants using the Watch Walk method [18]. To summarise, this method captures wrist-worn acceleration signals to identify walking patterns based on hand movements (e.g., arm-swing, hands in pockets) using machine learning with Support Vector Machine (SVM) classification. We examined 19 daily-life walk- and sleep-related biomarkers, including walking quality and quantity quantified through frequency-domain and peak detection analyses, and walking speed estimated using SVM regression. This study aimed to derive normative values for walking parameters for the cohort, and to compare these measures between men and women and across different age groups.

We have also expanded the methods section to provide a more in-depth description:

2.1. Development of the Watch Walk Digital Biomarkers

We recruited 101 participants aged 19 to 91 (mean 47, SD:18) (67% female) from Sydney, Australia and Hong Kong, China. All participants reported no osteoporosis, neurological impairment, recent fractures, recent joint replacements or existing conditions that restrict exercise tolerance. Written consent was obtained before data collection, and ethics approvals were obtained from the University of New South Wales and Hong Kong Caritas Institute of Higher Education Human Research Ethics Committees. Participants wore an accelerometer (AX3 data logger, Axivity Ltd, UK) over their dominant wrist and were videotaped during the data collection. They conducted thirty minutes of daily activities and walked on an electronic walkway (GAITRite, CIR System Inc, USA) at slow, usual and fast paces with six different hand positions, including walking (1) with arm-swing, (2) with hands in pockets, (3) with hands held static in front of the body (texting), (4) with the dominant hand held next to the ear (phonecall), (5) while carrying a bag over the shoulder, and (6) while carrying a briefcase/grocery bag.

We referenced the video recordings for the synchronization and annotation of the accelerometric signals and ground-truth electronic walkway measurements, including walking speed and step time. We derived the sample level Euclidean norm from the annotated accelerometric signals and filtered it with a Butterworth bandpass filter (5th order, 0.25 Hz low cutoff frequency, 2.5 Hz high cutoff frequency, and 100 Hz sampling frequency). The filtered signals were then segmented into non-overlapping 4-second windows. Fifty-four features (supplementary table 3) were extracted for training and validating a two-stage multiclass support vector machine (SVM) classification model with ten k-fold validation at the participant level. In the first stage, the 4-second window was first classified into “walks with arm-swing”, “other complex walking”, “running”, “stationary”, “unspecified arms activities while sitting/ standing”, or “unspecified arms activities while walking”, with the class weights set to [50, 25, 10, 5, 5, 5] to focus on walking. The second stage further classifies windows in the “other complex walking” into one of the following: “walking with hands in pocket”, “walking with hands held stationary”, “walking with hand held next to the ear”, “walking with hand resting on the shoulder”, and “ walking with a briefcase/ grocery bag”. The SVM model was configured with a Radial Basis Function (RBF) kernel, with a kernel coefficient, gamma, of 1/54.

Subsequently, we identified their longest continuous walking bouts by counting the greatest number of consecutive walking windows. To address brief pauses within extended walking bouts, gaps meeting the three criteria below were bridged: (1) not exceeding 60 seconds, (2) not exceeding both the following and preceding consecutive walking window durations and (3) not exceeding one-fifth of either the following or preceding consecutive walking window durations. The proportion of walks ≥ 8 seconds and 60 seconds were calculated through the cumulative exposure of walking durations. Steps were identified by cross-checking the peak signal detection with auto-correlation coefficients within each window identified as walking. A corrective factor derived from the unspecified arm activities was added to the total step count.  We regrouped walks into episodes of eight steps each and calculated cadence by timing these episodes. We calculated the standard deviation of step times within each episode and derived step-time variability as the 95th percentile of the day. Step and stride regularity were derived as the normalized coefficients of the first and second peaks in the autocorrelation function and range between -1 to +1. Walking speed was estimated with SVM regression with ten k-fold validation (data partitioned at the participant level) by fitting (1) the participant’s sex, (2) the participant’s body height, (3) the median, (4) the interquartile range of the static-block-removed acceleration signal vector magnitude, (5) mean crude acceleration signal vector magnitude, (6) mean step time, and correlation coefficients between acceleration signals in the (7) x- and y- axes and (8) x- and z-axes in windows classified as walking with arm-swings. The 95th percentile and median were identified as maximal and usual walking speeds. Sleep duration and bedtime were extracted from the wrist accelerometer data based on the van Hees approach [19].

Round 2

Reviewer 1 Report

Comments and Suggestions for Authors

I read all the answers the paper's authors formulated to my questions. I believe that they responded reasonably and correctly to all the issues raised. For this reason, I agree with the publication of the work in your journal.

Author Response

We would like to thank the reviewer for their favorable and constructive feedback.

Reviewer 2 Report

Comments and Suggestions for Authors

Dear Authors,

The manuscript has been improved. However, there are still some shortcomings that reduce the scientific value of the study. There is a lot of information and comparisons that are not related to the purpose of the study.

In particular, an explanation is needed of the meaning of comparing the characteristics of the participants included and excluded from the study.

The results can be supplemented with correlations of numerous parameters that were recorded by wearable devices and presented in tables with walking speed. This will provide more complete information about the influence of different factors on physical activity.

The presentation of data in Table 2 should be adjusted, since there are different units of measurement: mean (SD) or median [IQR], n (%).

Revise the figures. Select pictures that reflect the results in paragraph 3.4 and transfer them to the main text. Move less significant figures (for example, Figure 1) to the appendices.

Consider transferring detailed information about your results from the caption to the main text.

My overall comment, the manuscript is not ready for publication in its present form, as it has low readability and needs to be edited.

Author Response

Reviewer 2:

Comment 1:

The manuscript has been improved. However, there are still some shortcomings that reduce the scientific value of the study.

Response 1:

We would like to thank the reviewer for their comments. Please find our point-to-point response below.

Comment 2:

There is a lot of information and comparisons that are not related to the purpose of the study.

In particular, an explanation is needed of the meaning of comparing the characteristics of the participants included and excluded from the study.

Response 2:

Table 1 aims to demonstrate possible selection bias and this study’s generalizability to the broader population. To improve readability, we have now excluded socioeconomic and health participant characteristics that have less relevance to walking and sleeping behaviours.

Comment 3:

The results can be supplemented with correlations of numerous parameters that were recorded by wearable devices and presented in tables with walking speed. This will provide more complete information about the influence of different factors on physical activity.

Response 3:

We have added a figure illustrating Pearson’s correlation between digital biomarkers as Supplementary Figure 4.

Comment 4:

The presentation of data in Table 2 should be adjusted, since there are different units of measurement: mean (SD) or median [IQR], n (%).

Response 4:

Thank you for your suggestion. We have revised Table 2 to place the “%” symbol in front of the variable instead of using “(%)” behind the variable. This change helps to avoid confusion for readers, as “(%)” could be misinterpreted as indicating both frequency and percentage, n (%), rather than just the unit. We have also denoted units after commas, instead of the previous brackets to avoid confusion. Additionally, we added footnotes, where † denotes mean (standard deviation) and ‡ denotes median [interquartile range]. Similar modifications have been applied to Supplementary Tables 1, 2 and 3 for consistency.

Comment 5:

Revise the figures. Select pictures that reflect the results in paragraph 3.4 and transfer them to the main text. Move less significant figures (for example, Figure 1) to the appendices.

Response 5:

We agree that the figures that reflect the results in paragraph 3.4 are important for the flow of reading. We have moved four related figures (Figure 5, 6, 7, 8) to the main text. We decided to keep Figures 1 and 2 in the main text, because they visualise the normative value which is critical to this paper as well.

Comment 6:

Consider transferring detailed information about your results from the caption to the main text.

Response 6:

We have moved part of the figure captions from Figures 3, 4, 7, and 8 to the main text in section 3.3.